# Prevalence of Polycystic Ovary Syndrome (PCOS) and Its Associated Risk Factors among Medical Students in Two Countries

**DOI:** 10.3390/ijerph21091165

**Published:** 2024-09-02

**Authors:** Gulam Saidunnisa Begum, Noor Alhuda Talal Almashaikhi, Maryam Yousuf Albalushi, Hajar Mohammed Alsalehi, Rayan Salih Alazawi, Bellary Kuruba Manjunatha Goud, Rajani Dube

**Affiliations:** 1Department of Biochemistry, College of Medicine and Health Sciences, National University of Science and Technology, Al Tareef, Sohar 321, Oman; gulambeguam@nu.edu.om; 2College of Medicine and Health Sciences, National University of Science and Technology, Al Tareef, Sohar 321, Oman; noor190266@nu.edu.om (N.A.T.A.); maryam180484@nu.edu.om (M.Y.A.); hajar180414@nu.edu.om (H.M.A.); rayan180330@nu.edu.om (R.S.A.); 3Department of Biochemistry, RAK College of Medical Sciences, RAK Medical and Health Sciences University, Ras Al Khaimah P.O. Box-11172, United Arab Emirates; manjunatha@rakmhsu.ac.ae; 4Department of Obstetrics and Gynecology, RAK College of Medical Sciences, RAK Medical and Health Sciences University, Ras Al Khaimah P.O. Box-11172, United Arab Emirates

**Keywords:** polycystic vary syndrome, PCOS in medical students, risk factors of PCOS, obesity and PCOS, complications

## Abstract

Introduction: PCOS, a common hormonal disorder in women of reproductive age, affects fertility and increases the risks of other diseases. Early detection, risk factor assessment, and intervention are crucial to prevent long-term complications. Materials and Methods: This study was conducted using a pre-validated questionnaire at two medical colleges in the UAE and Oman. The first study (UAE) results are already published. Here, we present the findings of the second study (Oman) and compare them. Results and Discussion: The prevalence of PCOS was 4.6% (*n* = 7) in Oman and 27.6% (*n* = 69) in the UAE using the NIH criteria. The most common symptoms were irregular periods, acne, and thinning of hair. Students showed acne as the most prevalent symptom of clinical hyperandrogenism. Omani students showed significantly more acne [70.1% (*n* = 108) vs. 41.6% (*n* = 104)], while Emirati students showed a higher prevalence of hirsutism [32% (*n* = 80) vs. 23.3% (*n* = 36)]. A higher number of students had irregular periods 30.8% (77/150) in the UAE, although the difference was not statistically significant. The prevalence of PCOS was significantly higher in Emirati medical students than in Omani students (*p* < 0.05). The prevalence was also lower among medical students in Oman compared to an unselected population, reported by a study that included all consecutive women between 12 and 45 years of age attending a hospital. An increased trend in unhealthy lifestyle practices was observed in the recent study. Obesity was a strong predictor of PCOS symptoms across the populations in both countries (*p* < 0.05). Conclusion: The prevalence of PCOS and clinical signs of hyperandrogenism vary significantly between countries in the MENA region. There is a need to identify specific risk factors associated with PCOS in different populations, explore the genetic basis, and undertake collaborative efforts among healthcare professionals from various disciplines to raise awareness about PCOS and its associated risks.

## 1. Introduction

Polycystic ovary syndrome (PCOS) is one of the most prevalent endocrinological problems impacting women in their reproductive years. It presents a triad of symptoms including elevated androgens, irregular menstruation, and infertility. PCOS may also result in additional health complications, including type 2 diabetes and cardiovascular issues. In order to avoid lifetime consequences, early screening, lifestyle modification, and timely intervention are required. PCOS is estimated to affect around 6–7% of the global population [1]. The World Health Organization estimates that PCOS affects 8–13% of women of reproductive age and that more than half of the cases are undiagnosed [2]. Most epidemiological studies on PCOS have been conducted in developed countries, with only limited information available on its burden in the Middle East and North Africa (MENA) region [3,4,5]. In a study conducted among health sciences students in South India, the prevalence of PCOS was reported to be of 9.13% [6]. Likewise, research conducted among Omani women between the ages of 12 and 45 years including all consecutive women attending the hospital (unselected population) indicated that 7% of them were diagnosed with PCOS. However, it did not explore the specific age-wise prevalence or specified it in the student population [7]. The latest study in the United Arab Emirates (UAE) shows a self-reported prevalence of 25.9% with a significantly increased association of chronic disease later in life [8]. A systematic review demonstrated the prevalence of PCOS in adolescents by NIH criteria to be of 3.39% (95% CI: 0.28–9.54%). However, it did not include the population from the UAE or Oman [9]. A recent report from the MENA region in 2019 shows a 37.9% increase in prevalence since 1990 [10]. It is suggested that this increased prevalence is likely due to changes in lifestyle, diet, and stress [11,12,13,14,15].

The pathophysiology of the development of PCOS is complex, and although an array of risk factors is identified, none have been proven as a causative factor. The common risk factors identified are family history of PCOS, lack of physical exercise or a sedentary lifestyle, specific dietary habits, and obesity [16,17,18]. Physical inactivity has been associated with symptoms that overlap with metabolic syndromes like obesity, hypertension, peripheral insulin resistance, PCOS, and dyslipidemia [19,20]. It has been shown that physical exercise can be used as an independent treatment for PCOS women to appraise all PCOS phenotypic characteristics [21,22].

There are several criteria used for the diagnosis of PCOS. The most common ones used are the National Institute of Health (NIH) and Rotterdam criteria [23,24]. The NIH criteria were used for PCOS diagnosis in this research and the details are mentioned under the inclusion criteria. The criteria were expanded in 2003 with the introduction of the Rotterdam criteria, considered the gold standard, which require the presence of two out of the following three classical features of PCOS in the absence of congenital adrenal hyperplasia, hyperprolactinemia, and androgen-secreting neoplasms [24]:Oligo- and/or anovulation (ANOVU);Signs of hyperandrogenism, whether clinical or biochemical (serum testosterone and DHEAS);Polycystic ovaries on ultrasonography (the presence of eight or more subcapsular follicular cysts measuring less than 10 mm and an increase in ovarian stroma could serve as a significant indicator of PCOS).

There are some unique features of the manifestation of PCOS in adolescents. The temporary decline in insulin sensitivity observed during puberty may be a critical moment in the development of IR and hyperinsulinemia in patients predisposed to developing PCOS [25]. The associated clinical findings of insulin resistance (IR) and hyperandrogenism usually begin to manifest during adolescence in females with a tendency to develop PCOS, and these may be exacerbated in the presence of obesity [26,27]. Features like irregular menses and acne are normally seen in adolescents and patients seek evaluation when the symptoms become worrisome and home treatments are no longer effective. These are typically of great concern to the adolescent patient who is in a period of life when a semblance of normalcy with peers is critically important [25]. Lifestyle changes, medications, and surgery are all considered effective treatment options for PCOS [11,12,13,14,15]. Lifestyle interventions during late adolescence are useful in the prevention of long-term complications.

There is limited available data regarding the prevalence and risk factors of PCOS among medical students in countries like the UAE and Oman [28]. The level of awareness of PCOS was also found to be insufficient in the UAE in a recent study [29]. A better understanding of the current prevalence in the region and studying the disease trend is important to inform policy decisions regarding the interventional strategies for the prevention of the disease in adolescents. This will help in the allocation of public health resources to minimize the long-term effects of the disease.

Hypothesis: We hypothesized that PCOS is prevalent among medical students in late adolescence due to stress and academic lifestyle in the UAE and Oman, but in different ways in the two countries. We further put forth the hypothesis that the trend in risk factors associated with PCOS is showing a rise.

Objective:

The main objective was to analyze the trend in PCOS prevalence and risk factors among female medical students of reproductive age in the MENA region by comparing data from a previously published study in the UAE (2016) and new data collected in Oman (2023).

## 2. Materials and Methods

This prospective, descriptive cross-sectional survey was conducted in the UAE and Oman following approval from the Ethics and Biosafety committee of both the College of Medicine and Health Sciences (COMHS), National University of Science and Technology, Sohar, Sultanate of Oman, and the RAK College of Medicine, RAK Medical and Health Sciences University (RAKMHSU), RAK, UAE. Upon obtaining their written informed consent via convenient sampling, female students aged between 18 and 21 years who expressed willingness to participate were enrolled in the study.

This study was conducted in two separate instances: the first included students from RAKMHSU in the UAE in 2016, and the second included students from Oman in 2023. Both the MD1 and MD2 batches were administered a pre-validated questionnaire during class sessions, at different times. It was ensured that all participants in the study completed the data collection form including the questionnaire. Individuals were then classified into PCOS and non-PCOS groups based on NIH criteria. The results of the first (UAE) study are already published, and the second study (Oman) is presented here. The trend of the disease was estimated by comparing the results of the studies regarding the prevalence of PCOS and associated risk factors.

### 2.1. Participants

#### 2.1.1. Inclusion/Exclusion Criteria

Inclusion Criteria

Female students aged 18–21 years studying at the College of Medicine and Health Sciences, Oman, and the RAK Medical and Health Sciences University, UAE.Students who volunteered to participate.PCOS diagnosed by NIH criteria including all of the following:
A menstrual cycle that extends beyond 35 days or occurs less than 8 times per year (oligomenorrhea), or the total absence of menstruation (amenorrhea).Clinical signs of hyperandrogenism (HA) include symptoms of androgen excess such as acne, hirsutism, male-pattern (temporal) alopecia, and acanthosis nigricans (darkened, hyperpigmented skin thickening typically found at the nape of the neck and in the axilla), which serve as indicators of insulin resistance.Obesity is characterized by an elevated waist-to-hip ratio, often referred to as an “apple-shaped” body.

Exclusion Criteria

Students who are on medication for irregular periods.Students on oral hypoglycemic drugs.Those with underlying diseases of the pituitary, adrenal glands, and thyroid gland.Students who are pregnant.

#### 2.1.2. Sampling and Sample Size

Using an online calculator (calculator.net), at a 95% confidence level, and 5% margin of error, including all medical students between the ages of 18 and 21 years, the calculated sample size was 154 (in Oman). Recruitment was carried out consecutively by convenient sampling until reaching the predetermined minimum sample size.

### 2.2. Data Collection

The pre-validated questionnaire contained questions on risk factors like dietary habits, fast food consumption, physical activity levels, family history of the condition, and the presence of PCOS symptoms or signs [17,18]. The data collection form contained details on demographics, anthropometric measurements (including height, weight, and waist circumference), and detailed responses to the questionnaire [18,22]. All participants in the study completed the data collection form including the questionnaire, and individuals were then classified into PCOS and non-PCOS groups based on NIH criteria.

### 2.3. Statistical Analysis

The Statistical Package for the Social Sciences (SPSS) software version 26 (International Business Machines Corporation, New York, NY, USA) was used for all statistical analyses. Descriptive statistics was used for demographic data. Univariate analysis was used for demography and anthropometric parameters of participants and expressed in mean ± SD and percentages. Bivariate analysis was used for the comparison of independent variables between the groups (MD1–MD2; With PCOS-–Without PCOS; and UAE–Oman) using the Chi-square test, and Fisher exact test. Results are statistically significant if *p* < 0.05 with a 95% confidence interval.

## 3. Results

The results of the first study have already been published [18]. A total of 154 female students participated in the second study. It was seen in the UAE study that very few clinical year students (Years 3, 4, 5) met the inclusion criteria and participated voluntarily, as they were already diagnosed and on treatment by then. Hence, we focused on students in late adolescence in MD 1 and MD 2. The mean age of the participants was 19.55 ± 0.08. Most students (MD 1, 72 out of 76; MD 2, 73 out of 78) were Omani nationals. All the participants satisfying NIH criteria (anovulation, signs of HA, and overweight) for PCOS were categorized under the PCOS group and the remaining participants were in the non-PCOS group. Seven participants (5.26% for MD1 and 3.84% for MD2) were diagnosed with PCOS [Table 1]. Conversely, the remaining 147 participants in the non-PCOS group exhibited either anovulation or signs of HA but did not meet the NIH criteria for a PCOS diagnosis. Additionally, participants without any discernible features were also categorized in the non-PCOS group. All participants diagnosed with PCOS were Omani nationals. The most common symptoms were acne, irregular periods, and scalp hair thinning. A waist circumference of more than 80 cm was seen among 22 participants [Table 1].

Among the signs of HA, the least common was excessive skin tags (13.14% of MD1; 8.9% of MD2). Approximately 27.6% (*n* = 21) of MD1 students and 19.2% (*n* = 15) of MD2 were having irregular periods, and the most common sign of HA was acne (69.7% of MD1; 70.4% of MD2) among the participants even without a diagnosis of PCOS [Figure 1].

Risk factors of PCOS identified by the study in the UAE were a positive family history of PCOS, high-calorie dietary habits, fast food consumption (>3 times/week), lack of physical exercise (≤3 times/week), being overweight or obese, having high blood pressure (BP) (systolic BP of >140 mmHg or diastolic BP of >90 mmHg or both), and a waist circumference >80 cm, which were also seen among students in Oman [Figure 2]. Twelve students (7.8%) had elevated systolic BP while 17 (11%) had elevated diastolic BP. Among the Omani students, increased waist circumference (>80 cm) and being obese were significantly associated with the diagnosis of PCOS [Table 2].

When the trend was compared in the adolescents regarding the prevalence and clinical features of PCOS among Emirati and Omani populations over the years, the mean age and relative composition of the population with regard to countries of the MENA region were similar. The results of the UAE study have already been published [18]. The menstrual irregularities like oligomenorrhea, amenorrhea, and hirsutism were similar in both populations (Table 3). The prevalence was significantly higher among students in the UAE than in Oman. While all PCOS cases were Omani in the second study, local Emirati students had a significantly higher prevalence of PCOS (9 out of 20) than non-Emirati students (*p* = 0.0442) in the UAE. The presentation of acne, unhealthy food habits, and a lack of exercise were seen significantly more in the recent study compared to the previous one. However, the rates of acanthosis, obesity, and waist circumference >80 cm were significantly lower in the Omani population.

## 4. Discussion

The global prevalence of PCOS is estimated to range from 6% to 7% [30]. In a study conducted among women in Oman, the prevalence was found to be of 7% [31]. In the recent study, the prevalence was found to be of 5.26 and 3.83% in MD1 and MD2, respectively, which is lower. The findings of the UAE study are in accordance with other studies showing a high prevalence of PCOS in the MENA region, but the PCOS was self-reported [8]. Medical students are often educated about the importance of healthy lifestyles, including regular exercise and balanced nutrition. This emphasis on health promotion and disease prevention may contribute to lower PCOS prevalence rates among this group. Our study findings align with a prevalence rate of 3.86% for PCOS among tenth-grade schoolgirls in the Guangzhou area, as determined by the NIH criteria [32]. A recent study shows an increase of around 67% in the age-specific prevalence of PCOS in late adolescents in the United States of America over a 12-year period [33]. This may be due to the fact that the earlier study did not specifically involve the UAE or Omani population, nor the late adolescents or medical university students [10].

Participants with a family history of PCOS were found to have a significantly elevated risk (RR 2.18 and 3.8) of developing PCOS compared to those without such familial history. Similar findings were reported in another study with a 3-fold increase of PCOS in the offspring of mothers with PCOS [34]. The study participants with a positive family history of PCOS were significantly more present in the PCOS groups, as observed among both studies at 21% and 17%. This is in line with other studies where 32% of the studied PCOS patients had a sister and 24% a mother with PCOS [35]. This indicates a transmission from both parents. Despite having a higher percentage of family history in Oman participants, the prevalence was significantly less. The genetic basis of PCOS has been debated. Our findings indicate that the gene expression may be complex, in accordance with other studies [36].

The metabolic effects of IR and hyperinsulinemia, with its steroidogenic and reproductive effects and adipokine production by subcutaneous and visceral fat, are thought to be factors responsible for PCOS in obesity [37]. The BMI of participants in both the PCOS and non-PCOS groups was computed and subsequently classified according to the World Health Organization criteria into normal (BMI 18.6–24.9 kg/m^2^), overweight (BMI 25–29.9 kg/m^2^), and obese (BMI 30–40 kg/m^2^). There was a statistically significant difference between underweight, normal BMI, overweight, and obese individuals with a higher risk of PCOS among overweight [RR = 4.9; 10] and obese participants [RR = 2.71]. This shows a strong correlation between increased BMI and PCOS symptoms even across different countries. A population-based study of girls in late adolescence (ages 15–19 years) suggested an incremental risk of development of PCOS (by NIH criteria) in overweight and obese individuals, and the prevalence of obesity was significantly higher in adolescents with PCOS compared with those without [38]. However, other studies have reported that its quantitative impact remains unclear [39].

A contributing risk factor for the development of metabolic diseases like PCOS is central obesity, marked by an increased waist circumference [40,41]. There was a statistically significant difference between individuals with central obesity characterized by increased waist circumference of more than 80 cm [RR = 28.8, 17.5] with higher risk of PCOS among participants with central obesity. Our study agrees with a recent systematic review, identifying abdominal obesity to be a major contributor to the development of PCOS in adolescents with approximately 50% of girls with PCOS having central obesity [41]. This condition often arises from excessive consumption of fast food and irregular eating habits, leading to significant fluctuations in blood glucose levels and hormonal imbalances, thereby increasing the risk of developing PCOS [42,43]. Increased waist circumference and BMI result in increased inflammatory markers, which cause IR, dyslipidemia, and cardiometabolic syndrome. Hyperinsulinemia stimulates ovarian thecal cells for androgen production, resulting in signs of HA. Lack of physical exercise was considered a risk factor [RR = 1.9 and 2.4] among the study participants. There is a correlation between excess body weight and lack of physical exercise, which contribute to IR.

When we looked into the signs of clinical HA, acne had the highest prevalence in Oman. The second most common sign was hair thinning followed by excess body hair, irregular period, dark patches, and skin tags. When compared with Emirati students, Omani students had a significantly higher prevalence of acne, even without PCOS, and a lower prevalence of hirsutism (although not statistically significant). This shows the diversity in manifestations of HA in different populations. It further highlights the need for the inclusion of more than one feature of HA for the diagnosis of PCOS in adolescent females with evidence from biochemical tests where possible. It is also important to highlight that the medical students’ lifestyle could be a cause of HA due to lifestyle changes and stress. Academic life may be overwhelming, which has an influence on the lack of exercise and poor diet, hence causing stress and anxiety.

Our study shows an increased fast food consumption among the students. This can directly or indirectly lead to metabolic diseases. According to our results, the diet of the individual had an impact on the development of PCOS. Unhealthy diets with high-sugar foods cause intestinal flora imbalance, which triggers chronic inflammation, IR, and features of HA. The trend demonstrated in this study is in accordance with earlier findings of increasing unhealthy lifestyles like consumption of fast food and lack of exercise, but with contrasting prevalence [33]. Despite the increase in the unhealthy habits (food and exercise) trend, the prevalence of PCOS is significantly lower in the Omani population. This shows a strong correlation between BMI and symptoms of PCOS, the role of genetic variations, and the multifactorial nature of the condition. Systolic blood pressure of more than 140 mm of Hg [RR = 2.7, 5.5] and diastolic blood pressure of more than 90 mm of Hg [RR = 4.4, 2.7] were considered as risk factors in the students of both years in Oman. It was also noted that obesity exacerbates the severity of hyperinsulinemia in women with PCOS. Hence, the first line of management for PCOS is recommended to be lifestyle changes [44]. Physical exercise, nutritional education, and behavioral guidance should be cornerstones of PCOS management and promoted right from adolescence to prevent long-term complications [42,43,44,45].

There are a few limitations in this study. The diagnosis of PCOS is not corroborated by an ultrasonographic assessment of polycystic ovaries, and the comparison groups belong to different nationalities, although both are in the MENA region at identical age groups. Due to the nature of the descriptive cross-sectional study design employed, only the potential risk factors could be quantified, without establishing a definitive link between these factors and PCOS. Also, the number of students with PCOS was small in Oman for robust statistical correlations. Further studies with more representative samples, longitudinal designs, and more precise measurements are needed to validate these findings and provide more definitive insights into the PCOS risk factors. Due to the fact that some of the variables (like exercise habits and fast food consumption frequency) are self-reported, there may be inaccuracies. However, strengths include the involvement of a similar population (of medical university students), using the same criteria for diagnosis, and an attempt to assess the trend in the disease as well as the risk factors in both countries over the years. Also, our future research will address the trend in both populations individually to analyze specific factors.

## 5. Conclusions

The risk factors associated with PCOS among our study participants included family history, fast food consumption habits, and obesity. The prevalence of PCOS varies significantly between countries in the MENA region. There is a need to identify specific risk factors associated with PCOS in different populations, as the likelihood of developing PCOS rises with the presence of one or more of these risk factors, including genetic predispositions. Many of these predisposing factors are modifiable, suggesting that vigilant monitoring and appropriate corrective measures could aid in both the prevention and effective management of PCOS. Addressing these factors is crucial for reducing the risk of long-term complications such as type 2 diabetes and cardiovascular disease. This necessitates collaborative efforts among healthcare professionals from various disciplines to raise awareness about PCOS and its associated risks among the public. By effectively managing and identifying predisposing factors, the onset of the condition can be delayed, and its management optimized.

In future research endeavors, uniform criteria for the diagnosis of PCOS need to be adopted globally to study the dynamics of the disease, and specific genetic aspects of PCOS need to be explored for the development of targeted preventive strategies.

## Figures and Tables

**Figure 1 ijerph-21-01165-f001:**
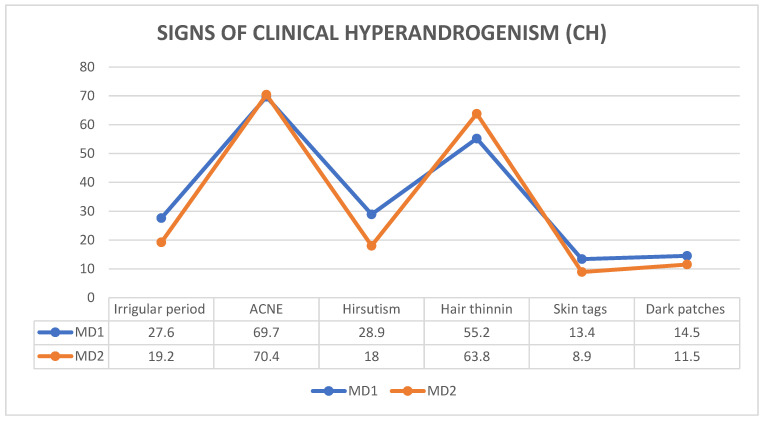
Signs of clinical hyperandrogenism among Omani students.

**Figure 2 ijerph-21-01165-f002:**
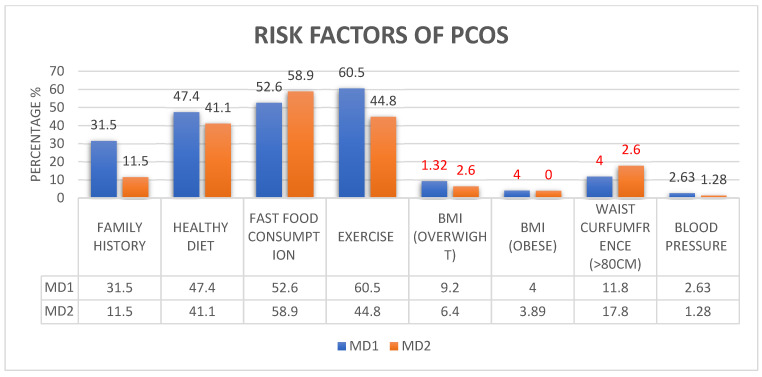
Details of risk factors of PCOS among MD1 and MD2 students. Risk factors of PCOS are a positive family history of PCOS, high-calorie dietary habits, fast food consumption (>3 times/week), lack of physical exercise (≤3 times/week), being overweight or obese, waist circumference > 80 cm.

**Table 1 ijerph-21-01165-t001:** Demography and clinical features of PCOS among participants in PCOS and non-PCOS groups in Oman.

Parameters	MD1; Age: 19.04 ± 0.08			MD2; Age: 19.98 ± 0.06 Years		
	PCOS (*n* = 4) (5.26%)	Non-PCOS (*n* = 72)		PCOS (*n* = 3) (3.84%)	Non-PCOS (*n* = 75)	
	With Symptoms	With Symptoms	Without Symptoms	With Symptoms	With Symptoms	Without Symptoms
	Number (%)	Number (%)	Number (%)	Number (%)	Number (%)	Number (%)
Nationality	Omani 4 (100%) Others—0	Omani 45 (66.1%) Others—2 (50%)	Omani 23 (33.8%) Others—2 (50%)	Omani 3 (100%) Others—0	Omani 48 (68.5%) Others—2 (40%)	Omani 22 (31.5%) Others—3 (60%)
Irregular period	4 (5.26%)	17 (22.36%)	55 (72.36%)	3 (3.8%)	12 (15.3%)	63 (80.7%)
Acne	4 (5.26%)	49 (64.47%)	23 (30.26%)	3 (3.7%)	52 (66.66%)	23 (29.48%)
Excess body hair	4 (5.26%)	18 (23.68%)	54 (71.05%)	3 (3.8%)	11 (14.1%)	64 (82.0%)
Hair thinning	4 (5.25%)	40 (52.63%)	32 (42.1%)	3 (3.8%)	39 (50%)	36 (46.15%)
Excess skin tags	4 (5.25%)	6 (7.89%)	66 (86.84%)	3 (3.8%)	4 (5.1%)	71 (91.0%)
Dark patches	4 (5.26%)	7 (9.21%)	65 (85.52%)	3 (3.8%)	6 (7.69%)	69 (88.46%)
Obese	4 (5.26%)	3 (3.94%)	69 (90.78%)	2 (2.5%)	3 (3.9%)	73 (93.5%)
Waist circumference > 80 cm	3 (3.94%)	6 (7.88%)	67 (88.15%)	2 (2.5%)	11 (15.3%)	65 (83.3%)

**Table 2 ijerph-21-01165-t002:** Details of risk factors of PCOS among students in Oman.

		MD 1 (*n* = 76)					MD 2 (*n* = 78)				
Variable	Category	PCOS (*n*)		Students (*n*)	RR (95% CI)	*p*-Value	PCOS (*n*)		Students (*n*)	RR (95% CI)	*p*-Value
		Yes	No				Yes	No			
**Family history of PCOS**	Yes	2	22	24	2.184 (0.324 to 14.477)	0.424	1	8	9	3.833 (0.385 to 38.150)	0.251
**Diet**	Unhealthy	12	28	40	0.83 (0.124 to 5.560)	0.850	3	44	47	0.214 (0.011 to 4.010)	0.302
**Physical activity**	No	1	29	30	1.956 (0.213 to 17.942)	0.552	1	42	43	2.457 (0.232 to 25.988)	0.455
**BMI**	Overweight	1	1	2	4.933 (0.807 to 30.156)	0.084 *	2	11	13	10.0 (0.977 to 102.314)	0.050 *
	Underweight	5	4	9			1	0	1		
	Obese	3	2	5	0.081 (0.013 to 0.478)	0.005 *	0	3	3	2.714 (0.167 to 44.078)	0.482
	Normal	1	59	60			3	58	61		
**Waist circumference**	>80 cm	3	5	8	28.875 (3.387 to 246.135)	0.002 *	2	6	8	17.500 (1.778 to 172.178)	0.014 *
**Blood pressure**	Systolic > 140 mmHg	0	2	2	2.777 (0.189 to 40.686)	0.455	0	1	1	5.571 (0.409 to 75.746)	0.197
	Diastolic > 90 mmHg	2	12	14	4.4286 0.6810 to 28.7980	0.119 *	0	3	3	2.714 (0.167 to 44.078)	0.482

* Significant.

**Table 3 ijerph-21-01165-t003:** Trends in the prevalence of PCOS and associated risk factors.

	Parameters	Oman (*n* = 154) N (%)	UAE * (*n* = 250) N (%)	Comments
1.	Age in years	19.55 ± 0.08	19.76 ± 1. 68	*p* > 0.05
2.	Nationality	Omani—145 Others—9	Emiratis—20 MENA region—83 SA countries—81 Others—66	
3.	Year of Study	Year 1 and 2	Year 1 and 2	
4.	Prevalence of PCOS	7 (4.6)	69 (27.6)	*p* < 0.001
5.	Locals in PCOS cases at each site (*n*) (RR; 95%CI)	7/145 (-)	9/20 (1.725; 1.0142 to 2.9339)	*p* = 0.0442
6.	Menstrual irregularities	36 (23.3)	77 (30.8)	*p* = 0.1064
7.	Hirsutism	36 (23.3)	80 (32)	*p* = 0.0627
8.	Acne	108 (70.1)	104 (41.6)	*p* < 0.001
9.	Acanthosis	20 (13)	79 (31.6)	*p* < 0.001
10.	Overweight and obese	6 (3.89)	79 (31.6)	*p* < 0.001
11.	Waist circumference >80 cm	16 (10.38)	94 (37.6)	*p* < 0.001
12.	Family history	33 (21.4)	42 (16.8)	*p* = 0.2451
13.	Fast food diet (>3 times/week)	87 (56.4)	20 (8)	*p* < 0.001
14.	Exercise habits ≤3 times/week	73 (47.4)	221 (88.4)	*p* < 0.001

* The data from the Phase 1 study, already published [18], SA = South Asian, MENA = Middle East and North Africa other than UAE locals.

## Data Availability

The raw data for Omani study are available with the corresponding author and can be obtained with reasonable request. All data relevant to the UAE study are already published (referenced) and are included in the article. The data are not submitted to any repository.

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
