# Peer review of "Prevalence of Polycystic Ovary Syndrome (PCOS) and Its Associated Risk Factors among Medical Students in Two Countries"

_ijerph, 2024, doi:10.3390/ijerph21091165_

Round 1

Reviewer 1 Report

Comments and Suggestions for Authors

Prevalence of polycystic ovary syndrome (PCOS) and its associated risk factors among medical students in two countries

Comment 1: Kindly mention the type of paper it is. (Case study/ clinical trials study/meta-analysis etc etc)

Comment 2: The informed consent form (ICF), must provide as supplementary documents

Comment 3: According to Fig 2 graph, exercise shows highest risk factor (60.5%). Author must describe how it is interlinked with PCOS directly.

Comment 4: Female 103 students aged between 18 and 25, who expressed willingness to participate were enrolled 104 in the study (line 103, 104)-Author did not mention the students who are <18 years and > 25 years students. What about age associated risk factors? 

Comments on the Quality of English Language

Moderate English Language Editing required

Author Response

Dear Reviewer,

                     Thank you for your insightful suggestions. We agree with the suggested modifications and modified the manuscript accordingly. The manuscript is extensively modified. 

Comment 1: Kindly mention the type of paper it is. (Case study/ clinical trials study/meta-analysis etc etc)- Mentioned and highlighted

Comment 2: The informed consent form (ICF), must provide as supplementary documents- Attached

Comment 3: According to Fig 2 graph, exercise shows highest risk factor (60.5%). Author must describe how it is interlinked with PCOS directly. Figure 3 mentions the risk- Explained in the text and the footnote

 Comment 4: Female 103 students aged between 18 and 25, who expressed willingness to participate were enrolled 104 in the study (line 103, 104)-Author did not mention the students who are <18 years and > 25 years students. What about age associated risk factors? Explained and highlighted in the text

Comments on the Quality of English Language- Moderate English Language Editing required- The entire manuscript is thoroughly checked for language and grammar and modified.

Reviewer 2 Report

Comments and Suggestions for Authors

Dear authors,

Polycystic Ovary Syndrome (PCOS), a common hormonal disorder among women of reproductive age, affects fertility and increases the risks of other diseases. Early detection, risk factor assessment, and intervention are crucial to prevent long-term complications. I consider your work is important, but the article needs major revision. The methodology lacks statistical foundation, results are not well presented and discussed, and limitations are not referred.

Abstract:

At abstract lane (29-30) “The prevalence of PCOS was significantly higher in the UAE than in Oman” (Ho many are from UAE and OMAN?)  and “The prevalence is lower among medical students compared to the unselected population.” (It is not clear what is the unselected population, from OMAN or from UAE?).

The results are presented as percentage you should present the number (n) of participants also.

Methodology (study design)

I believe that the methodological (statistical) approach to this work was not the most appropriate. In my opinion, you should reconsider presenting the results differently. It is not clear why you divided the sample into MD1 and MD2, as this makes your sample more difficult to analyse statistically and was not the main objective of the study, which was to compare differences between the first and second phases and between Oman and the UAE regions. Additionally, it is unclear how many participants from Phase 1 and Phase 2 are from Oman and the UAE. There are no reference to the statistical tests used or how the values are categorized. Additionally, the group of participants is not well described. It is necessary to include a table with descriptive statistics that characterizes the participants across all these groups.

Results /Discussion

While you present the differences between phase I and phase II related factors well, they are not properly discussed and in conclusion section you also do not refer that. You should highlight possible lifestyle differences trying to explain the observed differences in the two phases and between regions.

Given these considerations and suggestions, please consider revising the data analysis section and focusing on addressing the initial questions and objectives. Additionally, please note that the number of participants does not allow for robust statistical analysis.

Thank you for considering these suggestions.

Best regards.

Author Response

Dear Reviewer,

                      We thank you for the insightful suggestions. We agree with the points raised and modified the manuscript accordingly.

Abstract:

  • At abstract lane (29-30) “The prevalence of PCOS was significantly higher in the UAE than in Oman” (Ho many are from UAE and OMAN?)  and “The prevalence is lower among medical students compared to the unselected population.” (It is not clear what is the unselected population, from OMAN or from UAE?). Dear Reviewer, we agree with the suggestion. It is now explained and highlighted
  • The results are presented as percentages you should present the number (n) of participants also. Modified and highlighted

Methodology (study design)

I believe that the methodological (statistical) approach to this work was not the most appropriate. In my opinion, you should reconsider presenting the results differently.

It is not clear why you divided the sample into MD1 and MD2, as this makes your sample more difficult to analyze statistically and was not the main objective of the study, which was to compare differences between the first and second phases and between Oman and the UAE regions. Dear reviewer, we agree with the comments. The objective was to find the prevalence in late adolescents, risk factors, trends, and comparison. As the results of Phase 1 have already been published, we are presenting the results of Phase 2 and comparing them with Phase 1 results. The same has been updated in the manuscript and highlighted.

Additionally, it is unclear how many participants from Phase 1 and Phase 2 are from Oman and the UAE. updated in the manuscript and highlighted.

There are no references to the statistical tests used or how the values are categorized. updated in the manuscript and highlighted.

Additionally, the group of participants is not well described. It is necessary to include a table with descriptive statistics that characterize the participants across all these groups. The groups were similar with respect to year of study, being medical students and age (mentioned in table 5). Other anthropometric variables, life style factors and clinical features were also compared in table 5. We have added the difference in nationality in table 5 for comparison and mentioned in the text. Please let us know if any other parameters are required.

Results /Discussion

While you present the differences between phase I and phase II related factors well, they are not properly discussed and in conclusion section you also do not refer that. You should highlight possible lifestyle differences trying to explain the observed differences in the two phases and between regions. Dear reviewer, we agree with the comments. We have updated the manuscript to include these points and highlighted.

Given these considerations and suggestions, please consider revising the data analysis section and focusing on addressing the initial questions and objectives. Additionally, please note that the number of participants does not allow for robust statistical analysis

Round 2

Reviewer 2 Report

Comments and Suggestions for Authors

Dear Authors,

Your article continues to have some of the weaknesses that I initially pointed out.

 Introduction

This text” The diagnostic criteria for PCOS established by the National Institute of Health (NIH) 79 rely on the following was used to diagnose the PCOS according……………[23] 80 . A menstrual cycle that extends beyond 35 days or occurs less than 8 times per 81 years (oligomenorrhea), or the total absence of menstruation (amenorrhea). 82 2. ……………………………………………………………….Polycystic ovaries on ultrasonography (The presence of eight or more subcapsular 96 follicular cysts measuring less than 10 mm and an increase in ovarian stroma could serve 97 as a significant indicator of PCOS” must be included at the Material and Methods section, resumed as inclusion criteria. At Introduction you can refer that there are several classifications….(with references) but not as detailed as that.

Objectives

Consider altering the defined objectives at the introduction section, choosing a main objective and specifying sample features. The first 3 defined objectives are redundant, consider describing follows:

-        The main objective was to analyze the trend in PCOS prevalence and risk factors among female medical students of reproductive age in the MENA region by comparing data from a previously published study in the UAE (2016) and new data collected in Oman (2023).

Material and Methods

The sentence (line 138): “The study was done in two phases. The first phase included students from RAKMHSU in 2016 and the second phase included students in 2021” is not correct and is misleading. Describing the study as being done in 'phases' suggests a longitudinal approach, which it isn't. A sentence to accurately reflect can be: “The study was conducted in two separate instances: the first included students from RAKMHSU in the UAE in 2016, and the second included students from Oman in 2023.” (the same correction must be done in all texts). When you describe the results, you also explain the differences between the two instances as if it were a longitudinal study (line 247): “The presentation of acne, unhealthy food habits, and lack of exercise has been significantly more in the recent study compared to the previous one.” And you must explain that interestingly is contradictory to the results because there was a higher prevalence in the first study.

Another discussed item: “Our study shows an increased trend in fast food consumption over the years “ (line 324) This is not correct. You are not comparing the same population over the years; instead, you are comparing two populations that could already have different eating habits during the same period of time.

Results/Discussion

In the Results and Discussion section, the prevalence of PCOS was reported as “4.6% (n=7) in Oman and 27.6% (n=69) in the UAE using the NIH criteria. In Oman, the prevalence was higher among first-year MD students (5.26%; 4/76) compared to second-year MD students (3.84%; 3/78). Additionally, approximately 27.6% (n=21) of MD1 students and 19.2% (n=15) of MD2 students reported having irregular periods”.

I am very sorry, but I cannot understand the point of separating first- and second-year medical students to analyze differences when the ages of the MD2 students (19.98±0.06 years) and MD1 students (19.04±0.08 years) are very similar. The first study of AUE population  did not categorized the sample by MD year of the participants, and the health conditions related to PCOS typically do not change significantly within a one-year period.

Please consider restricting the discussion of differences between MD1 and MD2 students to one or two paragraphs and avoid using the graphics in figures 2 and 3. Instead, focus on the main objective by analyzing the differences between the two regions.

Some limitations related with statistical analysis were already mentioned but not well discussed. Further studies with more representative samples, longitudinal designs, and more precise measurements are needed to validate these findings and provide more definitive insights into the PCOS risk factors. The fact that some of the variables are self-reported, there may be inaccuracies. This was referred for some variables but not well explored .

Consider changing the sentence: “…. While all PCOS cases were Omani in phase 2 of the study, local Emirati students had a significantly higher prevalence of PCOS than non-Emirati students (p=.0442) in phase 1 “ (lane 244). This is confusing because all participants in “ phase 2” were Omani, and no Emirati participants were included in the second study.

I believe that the work still needs some changes and improvement in the results and discussion sections.

The first work was very concise, but in this one, I find it a bit challenging to follow some of the points being discussed and understand the direction you are aiming for.

Author Response

Dear Reviewer,

                        Thank you so much for your insightful comments and suggestions. We agree with the points. We have modified the manuscript and highlighted them in the text in RED.

Dear Authors,

Your article continues to have some of the weaknesses that I initially pointed out.

 Introduction

This text” The diagnostic criteria for PCOS established by the National Institute of Health (NIH) rely on the following was used to diagnose the PCOS according……………[23]. A menstrual cycle that extends beyond 35 days or occurs less than 8 times per years (oligomenorrhea), or the total absence of menstruation (amenorrhea). ……………………………………………………………….Polycystic ovaries on ultrasonography (The presence of eight or more subcapsular follicular cysts measuring less than 10 mm and an increase in ovarian stroma could serve as a significant indicator of PCOS” must be included at the Material and Methods section, resumed as inclusion criteria. At Introduction you can refer that there are several classifications….(with references) but not as detailed as that.

-Agreed and modified as suggested

Objectives

Consider altering the defined objectives at the introduction section, choosing a main objective and specifying sample features. The first 3 defined objectives are redundant, consider describing follows:

-        The main objective was to analyze the trend in PCOS prevalence and risk factors among female medical students of reproductive age in the MENA region by comparing data from a previously published study in the UAE (2016) and new data collected in Oman (2023).

-Agreed and modified as suggested

Material and Methods

The sentence (line 138): “The study was done in two phases. The first phase included students from RAKMHSU in 2016 and the second phase included students in 2021” is not correct and is misleading. Describing the study as being done in 'phases' suggests a longitudinal approach, which it isn't. A sentence to accurately reflect can be: “The study was conducted in two separate instances: the first included students from RAKMHSU in the UAE in 2016, and the second included students from Oman in 2023.” (the same correction must be done in all texts). -Agreed and modified as suggested throughout the text

When you describe the results, you also explain the differences between the two instances as if it were a longitudinal study (line 247): “The presentation of acne, unhealthy food habits, and lack of exercise has been significantly more in the recent study compared to the previous one.” And you must explain that interestingly is contradictory to the results because there was a higher prevalence in the first study.

-Agreed and explained

Another discussed item: “Our study shows an increased trend in fast food consumption over the years “ (line 324) This is not correct. You are not comparing the same population over the years; instead, you are comparing two populations that could already have different eating habits during the same period of time.

-Agreed and modified

Results/Discussion

In the Results and Discussion section, the prevalence of PCOS was reported as “4.6% (n=7) in Oman and 27.6% (n=69) in the UAE using the NIH criteria. In Oman, the prevalence was higher among first-year MD students (5.26%; 4/76) compared to second-year MD students (3.84%; 3/78). Additionally, approximately 27.6% (n=21) of MD1 students and 19.2% (n=15) of MD2 students reported having irregular periods”.

I am very sorry, but I cannot understand the point of separating first- and second-year medical students to analyze differences when the ages of the MD2 students (19.98±0.06 years) and MD1 students (19.04±0.08 years) are very similar. The first study of AUE population  did not categorized the sample by MD year of the participants, and the health conditions related to PCOS typically do not change significantly within a one-year period.Please consider restricting the discussion of differences between MD1 and MD2 students to one or two paragraphs and avoid using the graphics in figures 2 and 3. Instead, focus on the main objective by analyzing the differences between the two regions.

-Agreed. We have significantly restricted the comparison between the MD year 1 and 2 students and revised the text throughout the manuscript. We have removed one figure and 2 tables exclusively discussing MD1 and MD 2 students. We request to keep two figures as we are discussing the prevalence of risk factors in Omani students and also comparing and contrasting the signs of hyperandrogenism in the two population. Hope you will consider this request and allow us to use these two figures. 

 Some limitations related with statistical analysis were already mentioned but not well discussed. Further studies with more representative samples, longitudinal designs, and more precise measurements are needed to validate these findings and provide more definitive insights into the PCOS risk factors. The fact that some of the variables are self-reported, there may be inaccuracies. This was referred for some variables but not well explored. -Agreed and modified

Consider changing the sentence: “…. While all PCOS cases were Omani in phase 2 of the study, local Emirati students had a significantly higher prevalence of PCOS than non-Emirati students (p=.0442) in phase 1 “ (lane 244). This is confusing because all participants in “ phase 2” were Omani, and no Emirati participants were included in the second study. Agreed. Although both UAE and Oman are included in the MENA region, the population dynamics are different. While most of the university students in Oman are locals, the general population in UAE is mixed. The local Emirati students are the minority. Also, no Omani students are studying at RAKMHSU, and vice versa. Hence, we have compared the prevalence of PCOS among Emirati students only with those of Omani students. However, we are willing to remove the sentence if required.     

I believe that the work still needs some changes and improvement in the results and discussion sections.

 The first work was very concise, but in this one, I find it a bit challenging to follow some of the points being discussed and understand the direction you are aiming for. We hope this version is better.
